

# Weekly-derived top-down VOC fluxes over Europe from TROPOMI HCHO data in 2018–2021

Glenn-Michael Oomen[1], Jean-François Müller[1], Trissevgeni Stavrakou[1], Isabelle De Smedt[1], Thomas Blumenstock[2], Rigel Kivi[3], Maria Makarova[4], Mathias Palm[5], Amelie Röhling[2], Yao Té[6], Corinne Vigouroux[1], Martina M. Friedrich[1], Udo Frieß[7], François Hendrick[1], Alexis Merlaud[1], Ankie Piters[8], Andreas Richter[5], Michel Van Roozendael[1], and Thomas Wagner[9]

[1]Royal Belgian Institute for Space Aeronomy (BIRA-IASB), Brussels, Belgium
[2]Karlsruhe Institute of Technology (KIT), IMK-ASF, Karlsruhe, Germany
[3]Finnish Meteorological Institute (FMI), Sodankylä, Finland
[4]Saint Petersburg State University, Atmospheric Physics Department, St. Petersburg, Russia
[5]Institute of Environmental Physics, University of Bremen, Bremen, Germany
[6]LERMA-IPSL, Sorbonne Université, CNRS, Observatoire de Paris, PSL Université, 75005 Paris, France
[7]Institute of Environmental Physics, Heidelberg University, Heidelberg, Germany
[8]Royal Netherlands Meteorological Institute (KNMI), De Bilt, the Netherlands
[9]Max Planck Institute for Chemistry (MPI-C), Mainz, Germany

**Correspondence:** Glenn-Michael Oomen (glenn-michael.oomen@aeronomie.be)

**Abstract.** Volatile organic compounds (VOCs) are key precursors of particulate matter and tropospheric ozone. Although the terrestrial biosphere is by far the largest source of VOCs into the atmosphere, the emissions of biogenic VOCs remain poorly constrained at regional scale. In this work, we derive top-down biogenic emissions over Europe using weekly-averaged TROPOMI formaldehyde (HCHO) data from 2018 to 2021. The systematic bias of the TROPOMI HCHO columns is char-
acterized and corrected for based on comparisons with FTIR data at seven European stations. The top-down fluxes of biogenic, pyrogenic, and anthropogenic VOC sources are optimized using an inversion framework based on the MAGRITTEv1.1 chemistry transport model and its adjoint. The inversion leads to strongly increased isoprene emissions with respect to the MEGAN-MOHYCAN inventory over the model domain (from 8.1 to 18.5 $\mathrm{Tg\,yr^{-1}}$) which is driven by the high observed TROPOMI HCHO columns in southern Europe. The impact of the inversion on biomass burning VOCs ($+13\%$) and anthro-
pogenic VOCs ($-17\%$) is moderate. An evaluation of the optimized HCHO distribution against ground-based remote sensing (FTIR and MAX-DOAS) and in situ data provides generally improved agreement at stations below about $50°$ N, but indicates overestimated emissions in northern Scandinavia. Sensitivity inversions show that the top-down emissions are robust with respect to changes in the inversion settings and in the model chemical mechanism. However, the top-down emissions are very sensitive to the bias correction of the observed columns. Furthermore, the use of different a priori emissions has a significant
impact on the inversion results due to large differences among bottom-up inventories. In regions with variable meteorology, there is strong week-to-week variability in the observed HCHO columns. The top-down emissions, which are optimized at weekly increments, have a much improved capability of representing these large fluctuations than an inversion using monthly increments.



## 1   Introduction

The emissions of non-methane volatile organic compounds (NMVOCs) play an important role for both air quality and climate. They are precursors of secondary organic aerosols (SOAs, Claeys et al., 2004; Hallquist et al., 2009) and also impact tropospheric ozone levels through their interaction with $NO_x$ (Houweling et al., 1998; Churkina et al., 2017). Globally, the largest source of NMVOCs is of biogenic origin, making up about 85% of the total emissions as compared to 12% and 3% for anthropogenic and pyrogenic emissions, respectively (Stavrakou et al., 2018; Granier et al., 2019). Among the biogenic volatile

organic compounds (BVOCs), isoprene ($C_5H_8$) makes up about half of the global emissions, making it the most important NMVOC (Guenther et al., 2012; Sindelarova et al., 2014). Even though isoprene emission models agree on several important features such as temperature and light density dependence, seasonality, and interannual variability, there are still large inconsistencies among different inventories regarding the annual global emission of isoprene, ranging from $\sim 300\,\mathrm{Tg\,yr^{-1}}$ to $\sim 600\,\mathrm{Tg\,yr^{-1}}$ (Sindelarova et al., 2022). The main drivers of these discrepancies come from differences in the climate inputs,

the emission factors, and the vegetation fields entering the model (Arneth et al., 2011).

Isoprene has a strong influence on tropospheric chemistry due to its large emissions and its fast photochemical sink, mainly through its reaction with OH radicals. The latter leads to the formation of various organic compounds through a long cascade of chemical reactions (Wennberg et al., 2018; Müller et al., 2019), some of which remain poorly characterized, especially in remote environments characterized by low levels of nitrogen oxides ($NO_x$, defined as $NO+NO_2$). Isoprene has an average

lifetime of $\sim 1$ hour in the troposphere (Bates and Jacob, 2019). One of its highest-yield oxidation products is formaldehyde (HCHO), a compound that can be readily measured using both ground- and space-based detectors. The main source of HCHO is the oxidation of VOCs, whereas direct emission provides only a small contribution. Methane oxidation accounts for approximately 60% of the global HCHO budget (Stavrakou et al., 2009), but due to its long lifetime, it functions as a background source. Indeed, short-lived VOCs, such as isoprene and monoterpenes, have a larger impact on the spatial distribution of

HCHO. Additionally, HCHO is also relatively quickly destroyed through photolysis and reaction with OH, and has a lifetime of approximately 5 hours (Stavrakou et al., 2015). Transport processes are relatively unimportant for its measured distribution because of its short lifetime and of the rapid chemical processing leading to HCHO production (except at night, during winter, or in $NO_x$-poor environments, see Marais et al. (2012)). Consequently, HCHO is an excellent tracer and its total columns can be directly linked to the emissions of reactive NMVOCs.

Since the advent of satellite HCHO data products, many studies have been conducted on the derivation of top-down biogenic emissions in an inversion framework. This includes studies performed using satellite HCHO data from GOME (e.g. Palmer et al., 2006), SCIAMACHY (e.g. Stavrakou et al., 2009), OMI (e.g. Millet et al., 2008; Stavrakou et al., 2018), and GOME-2 (e.g. Stavrakou et al., 2015). HCHO is a trace gas with a low optical density in the atmosphere and its absorption features in the ultraviolet (UV) are heavily blended with BrO and $O_3$ lines. Consequently, HCHO vertical columns from satellite

measurements are relatively noisy and substantial spatial and temporal averaging is needed in order to derive reliable top-down emissions.



In this study, we use the HCHO product of the Tropospheric Ozone Monitoring Instrument (TROPOMI, De Smedt et al., 2018) mounted on the Sentinel-5P (S5P) satellite as observational constraints in the top-down inversion. Due to its unprecedented spatial resolution and high signal-to-noise ratio, we use weekly averages of the data at $0.5° \times 0.5°$ spatial resolution, whereas previous studies used averages at monthly or longer timescales. The emissions of biogenic VOCs like isoprene being strongly temperature dependent, HCHO volume mixing ratios can show strong temporal variability in the course of one month. We can expect that limiting the extent of temporal averaging will significantly improve the constraints in the inversion.

We focus our study on the European domain as depicted in Fig. 1. This domain contributes only a small amount to the global biogenic VOC budget, since the highest isoprene emissions take place in warmer, more densely vegetated regions such as tropical forests. Nonetheless, the quantification of the BVOC emissions in Europe is important, since the high levels of anthropogenic $NO_x$ and sulfate particles make those compounds more prone to the production of tropospheric ozone (Seinfeld and Pandis, 2016) and secondary organic aerosols (Schwantes et al., 2019; Bryant et al., 2020). Furthermore, the high amount of validation data available in the domain, as is shown in Fig. 1, makes this region an excellent target for our inversion study. Previous top-down VOC emission studies over Europe using HCHO data include Dufour et al. (2009) and Curci et al. (2010), which used the CHIMERE model in combination with SCIAMACHY and OMI observations, respectively. Their inversion studies resulted in slightly reduced isoprene emissions with respect to the a priori estimates obtained from the MEGAN bottom-up inventory (Guenther et al., 1995, 2006). In a global scale inversion study by Bauwens et al. (2016) using OMI HCHO data and the IMAGESv2 model, a moderate increase of isoprene emissions was inferred in both eastern and western Europe.

The inversion of VOC emissions is sensitive to constraints from satellite observations. Consequently, it is important to carefully validate the satellite measurements using ground-based and aircraft measurements. Several validation studies have shown that OMI and TROPOMI HCHO columns are too low over source regions with respect to validation datasets (Zhu et al., 2020; Vigouroux et al., 2020; De Smedt et al., 2021). This bias was not accounted for in previous HCHO emission inversion studies. The origin of the bias is unknown, although some known difficulties in the HCHO retrieval include aerosol effects, uncertainties in the treatment of clouds, spectral interferences with BrO and ozone, and a coarse albedo climatology (Vigouroux et al., 2020). The bias can be corrected for based on a linear regression of the ground-based and satellite vertical columns. In this work, we derive a bias correction based on ground-based FTIR measurements at stations located in the European domain.

We describe the different types of data (ground-based and satellite) used in this work in Sect. 2. Section 3 describes the setup of the inversion, as well as the chemistry transport model MAGRITTEv1.1 around which the inversion framework is built. A detailed characterization of the biases of the TROPOMI HCHO data product is presented in Sect. 4 based on FTIR and MAX-DOAS data over Europe. Based on this analysis, a procedure for correcting the biases is proposed and used in the emission inversions. The inversion results are shown in Sect. 5 including a comparison of the top-down model with ground-based data. In Sect. 6, we provide an analysis of the emission uncertainties by means of sensitivity inversions. Finally, we present the conclusions in Sect. 7.



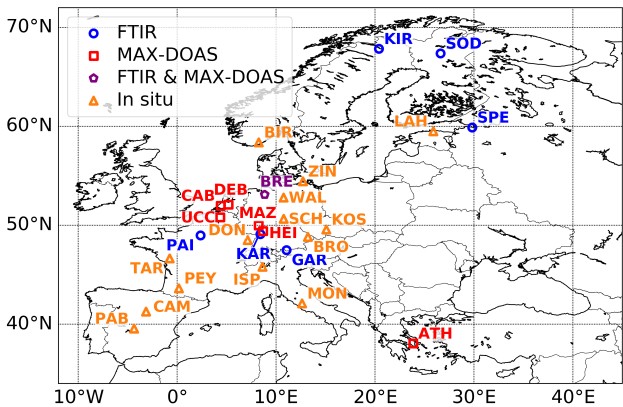

**Figure 1.** Locations of ground-based HCHO column stations and in situ HCHO measurement sites in Europe presented in Table 1.

## 2 HCHO Data

In this section, we provide an overview of the measurement datasets used in this work. TROPOMI HCHO column data are used to constrain the VOC emissions in the model (Sect. 2.1). We perform a careful validation of the satellite data using ground-based FTIR vertical columns from a range of stations across Europe (Sect. 2.2). We further validate the top-down results by including European MAX-DOAS stations in our ground-based remote sensing dataset (Sect. 2.3). Finally, in situ HCHO concentration data are also used to evaluate the model before and after inversion (Sect. 2.4). The full list of ground-based stations used in this work is shown in Table 1 and Fig. 1.

### 2.1 TROPOMI HCHO column densities

The S5P satellite has a nadir-viewing orientation in a low-Earth Sun-synchronous polar orbit. It has a daily equatorial overpass at 13:30 local time. The TROPOMI instrument, onboard the S5P platform, is an imaging spectrometer covering ultraviolet, visible, near-infrared, and short-wavelength infrared ranges. In the UV-VIS range, the initial spatial resolution of the instrument was $3.5 \times 7 \ \mathrm{km}^2$, which was further improved to $3.5 \times 5.5 \ \mathrm{km}^2$ after 6 August 2019.

The HCHO retrieval relies on the differential optical absorption spectroscopy (DOAS) method which allows for the isolation of narrow trace gas absorption features. The TROPOMI retrieval algorithm is based on the QA4ECV algorithm developed for its predecessor OMI (De Smedt et al., 2018). The first step is the derivation of the slant column density in the wavelength range 328.5–359 nm. The DOAS reference spectrum is updated daily with an average of Earth reflected radiances measured in a remote sector of the equatorial Pacific Ocean. The slant columns are converted into tropospheric vertical columns using altitude-dependent air mass factors, which are obtained from a look-up table calculated with the VLIDORT v2.6 radiative transfer model (Spurr et al., 2008), for a large number of atmospheric conditions depending on the observation angles, surface reflectivity, surface pressure and cloud properties. A priori vertical profiles are provided by the TM5-MP daily analysis on 34 vertical levels at the spatial resolution of 1° (Williams et al., 2017). The surface albedo is taken from the monthly OMI albedo





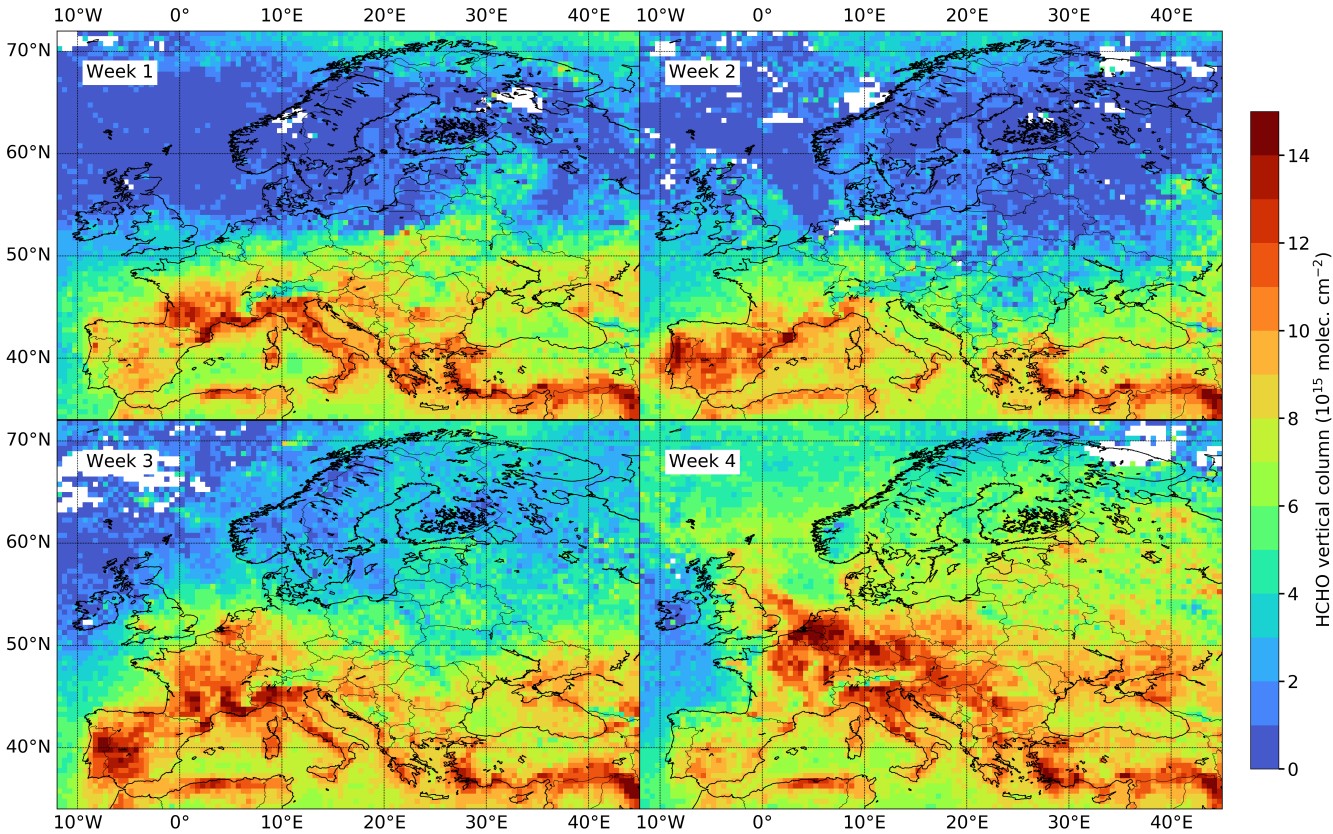

**Figure 2.** TROPOMI vertical HCHO columns over Europe for the different weeks of July 2019, gridded at $0.5° \times 0.5°$. White regions correspond to grid cells without valid data over the week (usually due to clouds or snow/ice). Week 1 corresponds to days 1 to 8, week 2 corresponds to days 9 to 16, week 3 corresponds to days 17 to 24, and week 4 corresponds to days 25 to 31.

climatology at $0.5°$ (minimum Lambertian equivalent reflectivity, LER, Kleipool et al., 2008). The final step of the algorithm consists of a background correction of the columns, using again the Pacific Ocean as reference sector. The background HCHO columns, mostly due to methane oxidation, are replaced by the TM5-MP model columns in the same sector (De Smedt et al., 2021).

        We use the operational offline level 2 data product (https://doi.org/10.5270/S5P-tjlxfd2) which provides HCHO tropospheric
column data together with uncertainty estimates, as well as a priori profiles and averaging kernels (De Smedt et al., 2018, 2021). The latter provide information on the vertical sensitivity of the satellite measurement, which is necessary to make meaningful comparisons with our model. We only select data with a quality value higher than 0.5, as recommended by the S5P HCHO product user manual (https://sentinel.esa.int/documents/247904/2474726/Sentinel-5P-Level-2-Product-User-Manual-Formaldehyde, last access: 21 August 2023). We use the clear-sky product, meaning that the cloud information is only used to
filter the satellite observations, and the cloud correction to the air mass factors is ignored (De Smedt et al., 2021). To ensure





that cloud contamination has a minimal effect on the total columns, we impose that the cloud fraction for each pixel is lower than 0.2, which is stricter than the standard cloud filter (fraction of 0.4 when using a quality value of 0.5).

Figure 2 shows the TROPOMI HCHO vertical columns over Europe for four weeks in July 2019, gridded spatially onto the model resolution ($0.5° \times 0.5°$). High temperatures and high levels of solar radiation generally favor biogenic emissions and
HCHO production from long-lived VOC precursors. On average, this leads to higher HCHO columns in southern Europe due to higher temperatures and radiation levels with decreasing latitude. Local minima generally correspond to mountain ranges, oceans, and arid or semi-arid areas. Meteorology plays an important role in the spatio-temporal variability of the observed columns (Fig. 2). Most notably, the high HCHO columns in central Europe in the last week of July 2019 are due to a record-breaking heat wave (Sousa et al., 2020). Weekly-averaged TROPOMI HCHO columns are used as observational constraints
in our inversion framework. At the resolution of $0.5°$, and using weekly-averaged columns, the precision of the TROPOMI HCHO columns is estimated to range between 1 and $3 \times 10^{15}$ molec. cm$^{-2}$, from background to continental conditions (De Smedt et al., 2021). The bias on the HCHO vertical columns is estimated to amount to $-30\%$ of the column for high HCHO levels, and to $30\%$ of the column for low HCHO levels (Vigouroux et al., 2020). A detailed derivation of the bias over the target domain is presented in Sect. 4.

## 2.2    FTIR column observations

The FTIR stations used in this study (Table 1) are part of the international networks TCCON (Total Carbon Column Observing Network, http://www.tccon.caltech.edu/, last access: 21 August 2023) and/or NDACC (Network for the Detection of Atmospheric Composition Change, https://www2.acom.ucar.edu/irwg, last access: 21 August 2023). They are recording remote sensing solar absorption measurements in the infrared domain, under clear-sky conditions, using high-resolution spectrometers
from the same manufacturer (Bruker 120/5HR). The retrievals of atmospheric total columns and low-resolution vertical profiles are performed using either one of the two algorithms available in the community (PROFITT9, Hase et al. 2006; and SFIT4, Pougatchev et al. 1995), both based on similar line-by-line forward models and optimal estimation methods (Rodgers, 2000) and therefore providing very consistent outputs.

To build a coherent dataset useful for satellite validation or model evaluation, the retrieval settings of HCHO have been
harmonized within the FTIR community (Vigouroux et al., 2018). This dataset has been used for the validation of TROPOMI (Vigouroux et al., 2020, updated in quarterly reports at https://mpc-vdaf.tropomi.eu, last access: 21 August 2023), GOME-2 (https://acsaf.org/valreps.php, last access: 21 August 2023) or more recently of OMPS (Kwon et al., 2023). Currently 28 stations provide HCHO column data, among which seven are in the European domain (Table 1; Fig. 1).

The details of the harmonized retrieval settings are documented in Vigouroux et al. (2018). In summary, the fitted spec-
tral signatures lie in the 3.6 μm region and belong to the $\nu_1$ and $\nu_5$ bands. The spectroscopic database is a compilation from G. Toon, the atm16 line list (http://mark4sun.jpl.nasa.gov/toon/linelist/linelist.html, last access: 21 August 2023), which corresponds to HITRAN 2012 (Rothman et al., 2013) for HCHO and is optimized for each of the interfering species. The Tikhonov regularization (Tikhonov, 1963) is used for constraining/optimizing the retrievals of low-resolution vertical HCHO profiles. The degrees of freedom of signal (DOFS, trace of the averaging kernel matrix) amount to only 1 to 1.4 for the stations used in



this study, implying that only total column information can be retrieved. The averaging kernel shape shows that the sensitivity is highest in the free troposphere (although still present at the surface).

The FTIR dataset is well characterized in terms of uncertainty budget, which is calculated following Rodgers (2000). The systematic uncertainty on the HCHO columns is about 13% (Vigouroux et al., 2018) and is dominated by the uncertainty on the spectroscopic parameters. The random uncertainties, dominated by the measurement noise, can be as low as $1.0 \times 10^{14}$ molec. cm$^{-2}$ (8%) for the pristine Eureka site and up to $5.3 \times 10^{14}$ molec. cm$^{-2}$ (7%) for the polluted Paris site. In addition, the FTIR data have significant random and systematic smoothing uncertainties (Vigouroux et al., 2018), but this component vanishes when the model profiles are smoothed using the FTIR averaging kernel matrix (Rodgers and Connor, 2003), as done in this study.

## 2.3  MAX-DOAS column observations

The MAX-DOAS (Multi-AXis Differential Optical Absorption Spectroscopy) technique (Hönninger et al., 2004; Platt and Stutz, 2008) involves recording UV-Visible spectra of the scattered skylight at different elevation angles between the horizon and the zenith. The different elevation angles yield different optical paths and thus different absorption strengths for trace gases present in the troposphere. By combining these multi-axis measurements with appropriate inversion schemes, it is possible to infer the vertical distribution and the integrated vertical column of several species absorbing in the UV-visible range, such as

NO$_2$, HCHO, or ozone. MAX-DOAS measurements also allow to quantify the absorption of the oxygen collision complex (O$_4$), which yields information on the tropospheric profile of aerosol extinction. The latter is crucial for the assessment of the optical path and the quantitative interpretation of the measured trace gas absorptions.

Table 1 presents the MAX-DOAS stations used in this work. They rely on various types of MAX-DOAS instruments differing in size, geometry, and signal-to-noise ratio, and are located in Germany, the Netherlands, Belgium, and Greece. Depending

on the spectral ranges of the spectrometers, the absorption of HCHO is either measured in the 324 to 359 nm or 336 to 359 nm spectral window, while O$_4$ is fitted in the 338 to 370 nm window. The consistency of the HCHO and O$_4$ absorption measurements has been demonstrated by regular intercomparison studies (e.g. Kreher et al., 2020). For stations that have multiple viewing directions, we treat the different directions equally.

Regarding the MAX-DOAS data analysis, this work uses the facilities developed within the FRM$_4$DOAS project (Fidu-

cial Reference Measurements for Ground-Based DOAS Air-Quality Observations, https://frm4doas.aeronomie.be, last access: 23 August 2023). FRM$_4$DOAS aims at harmonizing the data processing from MAX-DOAS instruments operated within the International Network for the Detection of Atmospheric Composition Change (NDACC), by incorporating existing retrieval algorithms into a fully traceable, automated, and quality-controlled processing chain.

The FRM$_4$DOAS evaluation starts with the production of differential slant column densities by applying the QDOAS anal-

ysis tool (Danckaert and Fayt, 2017). The QDOAS settings for FRM$_4$DOAS are described in Hendrick et al. (2018). The FRM$_4$DOAS system implements two MAX-DOAS retrieval algorithms: MAPA (Mainz Profile Algorithm, Beirle et al., 2019), which is based on a parameterization of the retrieval profile shape and a Monte-Carlo approach for the inversion, and MMF (Mexican MAX-DOAS Fit, Friedrich et al., 2019), an optimal estimation-based algorithm using the radiative transfer code



VLIDORT v2.7 (Spurr et al., 2008) as forward model. In this work, analogously to the official $NO_2$ product, only HCHO
MMF data consistent wth MAPA HCHO are retained. Both inversion algorithms have been extensively tested and validated
using synthetic (Frieß et al., 2019) and real data (Tirpitz et al., 2021; Karagkiozidis et al., 2022). Vigouroux et al. (2009)
quantified the uncertainties of individual MAX-DOAS measurements of HCHO VCDs. The systematic uncertainties mainly
originate from the HCHO cross-section and amounts to 9% of the HCHO VCD. The smoothing error (20%) dominates the
random error. After taking into account the retrieval noise (9%) and the errors on the forward model parameters (12%), this
leads to a total random error of around 25%.

## 2.4  In situ

Additional data to evaluate our model is provided by ground-based in situ measurement stations. We use data from the EMEP
network (Tørseth et al., 2012), available at the EBAS database (https://ebas.nilu.no, last access: 21 August 2023). The list of
stations with available HCHO concentration measurements is provided in Table 1. All in situ stations use high-performance
liquid chromatography (HPLC) to derive HCHO concentrations at daily or three-daily intervals.

For the majority of in situ stations in Table 1, measurements were taken before the time span covered in this study (i.e.,
before 2018). For those stations, we compute a climatological average of the measurements by averaging the data per month
across the different years. However, caution is required when comparing a climatological average to a recent model, since
HCHO concentrations are sensitive to meteorology and land-use changes.

## 3  Inversion methodology

### 3.1  Model description

The inversion method is built around the Model of Atmospheric composition at Global and Regional scales using Inversion
Techniques for Trace gas Emissions (MAGRITTEv1.1, Müller et al., 2019). This model is based on its predecessor IMAGES
(Müller and Stavrakou, 2005; Stavrakou et al., 2012; Bauwens et al., 2016) and simulates the atmospheric composition at
either global or regional scale. Its chemical mechanism includes 141 long-lived and 41 short-lived compounds. In particular,
it contains a state-of-the-art chemical mechanism for the atmospheric degradation of isoprene, based on the Leuven Isoprene
Mechanism (LIM1, Peeters et al., 2014) and includes updates from recent experimental studies (Wennberg et al., 2018; Berndt
et al., 2019). This chemical mechanism is thoroughly described in Müller et al. (2019). The only important update with respect
to the mechanism of Müller et al. (2019) concerns the isomerization rate of $\delta$-hydroxyperoxy radicals from isoprene, which
was based on a determination by the Caltech group (Wennberg et al., 2018). In this study, faster isomerization rates are used,
based on the LIM1 determination of the 1,6-H shift rate, combined with a five-fold enhancement of the $O_2$ addition and
elimination rates (also from LIM1). This enhanced rate was found necessary by Novelli et al. (2020) to reproduce their chamber
measurements of HOx radicals and isoprene oxidation products in dedicated oxidation experiments in the simulation chamber
SAPHIR.



Meteorological fields are obtained from the ECMWF ERA5 reanalysis data (Hersbach et al., 2020). The model is first run in its global configuration at a spatial resolution of $2° \times 2.5°$ (latitude, longitude). This simulation provides the boundary conditions for the MAGRITTEv1.1 model runs at $0.5° \times 0.5°$ over the European domain delimited by $34 - 72°$ N, $12°$ W $- 45°$ E. The model has 40 unequally-spaced vertical levels, extending from the surface up to a pressure of about 44 hPa.

Anthropogenic VOC emissions are obtained from the CAMS-GLOB-ANT inventory (Granier et al., 2019, version 4.3), biomass burning emissions from the Quick Fire Emissions Dataset (QFED) version 2.4 (Darmenov and da Silva, 2015) combined with emission factors from Andreae (2019), and biogenic emissions of isoprene, monoterpenes, and methanol are obtained from the Model of Emissions of Gases and Aerosols from Nature (MEGAN) coupled to the multi-layer canopy environment model (MOHYCAN, Müller et al., 2008; Guenther et al., 2012; Stavrakou et al., 2018). The anthropogenic VOC emissions display a slow declining trend, from 16.8 Tg VOC in 2018 to 16.6 Tg VOC in 2021. The annual biomass burning fluxes over the European domain amount to 1.6 and 1.7 Tg VOC in 2018 and 2021, whereas the fluxes are much higher in 2019 (2.5 Tg VOC) and 2020 (2.6 Tg VOC). The annual a priori isoprene fluxes amount to 8.2, 7.9, 7.8, and 8.4 Tg in 2018, 2019, 2020, and 2021, respectively.

The biogenic emission fluxes are dependent on a range of environmental and phenological factors. In the MEGAN model, the biogenic emission flux $F$ is calculated as

$$F = \epsilon \cdot C_{\mathrm{CE}} \cdot \gamma_{\mathrm{A}} \cdot \gamma_{\mathrm{CO_2}} \cdot \gamma_{\mathrm{SM}} \cdot \sum_{k=1}^{8} \left[ (\mathrm{LAI})_k \cdot (\gamma_{\mathrm{PT}}) \right], \tag{1}$$

where $\epsilon$ represents the standard emission rates obtained from the gridded distribution of Guenther et al. (2012), and $C_{\mathrm{CE}} = 0.52$ is an adjustment factor ensuring that $F$ equals $\epsilon$ in standard conditions (Müller et al., 2008). LAI is the leaf area index, and the different $\gamma$ factors describe the emission activity and are related to photon density and leaf temperature (PT), leaf age (A), $CO_2$ concentration, and soil moisture (SM) (see Guenther et al., 2006). In this work, we neglect the effects of drought ($\gamma_{\mathrm{SM}} = 1$) and we use the parameterization of Possell and Hewitt (2011) for $\gamma_{\mathrm{CO_2}}$. The summation in Eq. 1 is performed over 8 layers of the canopy, for which the MOHYCAN model calculates the direct and diffuse radiative fluxes and leaf temperature in each layer. For the above-canopy meteorological parameters, the ECMWF (European Centre for Medium-range Weather Forecasts) ERA5 reanalysis is used (Hersbach et al., 2020).

The model accounts for diurnal variations of the chemical compounds through correction factors on the chemical reaction rates, photolysis rates, convective fluxes and boundary layer mixing diffusivities computed via a full simulation of the diurnal cycle using a time step of 20 minutes (Stavrakou et al., 2009; Bauwens et al., 2016). Diurnal profile shapes calculated with the short time step are also used to estimate HCHO vertical columns at the overpass time of the TROPOMI instrument. The inversion is carried out with a time step of 24 hours, using daily-averaged chemical and dynamical parameters adjusted for the diurnal cycle using the pre-calculated correction factors. This setup provides an optimal balance between model accuracy and computational cost.



## 3.2  Inversion setup

The inversion aims at optimizing the emissions in the model in order to improve the comparison between modeled HCHO columns and TROPOMI observations. The emissions are optimized per emission category (biogenic, pyrogenic, and anthropogenic), per grid cell, and per week. We only use TROPOMI data between May and September, when HCHO columns are

highest, biogenic and pyrogenic emissions become important, and the photochemical lifetime of HCHO precursors is short, resulting in a minimal impact of dilution and transport of VOCs away from the emission areas. We always consider four weeks per month, which are defined as sets of (8, 7, 8, 7) or (8, 8, 8, 7) days for months with 30 days and 31 days, respectively. The inversions are performed independently for each year. We can write the top-down emissions $G(x,t)$ as

$$G(x,t) = \sum_{j=1}^{3} \phi_j(x,t) \exp\Big( f_j(x,t) \Big), \tag{2}$$

where $j$ denotes the emission category, $\phi_j$ are the a priori emission fluxes, $f_j$ are the emission parameters to be optimized, and $x$ and $t$ are the spatial coordinates (longitude and latitude) and time (week). For the biogenic component, the emission parameters are only applied to isoprene and monoterpenes, as they make up the majority of the total biogenic emissions. Furthermore, isoprene and monoterpenes are short-lived, and therefore have a strong impact on the HCHO distribution. The fluxes from a pixel and source category are only optimized when the a priori weekly value exceeds $10^{10}$ molec. cm$^{-2}$s$^{-1}$ at

least once throughout the year. The total number of parameters to be optimized by the inversion is of the order of $2 \times 10^5$ for each year.

The modeled HCHO columns are compared to the observed, bias-corrected HCHO columns of TROPOMI after application of the TROPOMI averaging kernels to the modeled profiles. We define the cost function $J$ by

$$J(\mathbf{f}) = \frac{1}{2}\Big[ \big(H(\mathbf{f}) - \mathbf{y}\big)^T \mathbf{E}^{-1} \big(H(\mathbf{f}) - \mathbf{y}\big) + \mathbf{f}^T \mathbf{B}^{-1} \mathbf{f} \Big], \tag{3}$$

where $\mathbf{f} = f_j(x,t)$ is a dimensionless vector consisting of the emission parameters of Eq. 2 to be optimized, $H(\mathbf{f})$ denotes the HCHO columns simulated by the model and smoothed by the TROPOMI averaging kernels, $\mathbf{y}$ are the bias-corrected TROPOMI HCHO columns, and $\mathbf{E}$ and $\mathbf{B}$ are the error covariance matrices of the HCHO columns and emission parameters, respectively (Müller and Stavrakou, 2005). The first term in Eq. 3 quantifies the difference between the modeled and observed HCHO columns, whereas the second term acts as regularization, preventing the optimized emissions from excessively deviating

from the a priori emissions. The error covariance matrix $\mathbf{E}$ is assumed to be diagonal (i.e., errors are uncorrelated) and its elements consist of the retrieval errors on the HCHO columns, to which a model error taken equal to $2 \times 10^{15}$ molec. cm$^{-2}$ is quadratically added. The matrix $\mathbf{B}$, on the other hand, has a diagonal component consisting of the square of the relative errors on the flux parameters $\mathbf{f}$ which are set equal to 0.9 (corresponding to an uncertainty factor of 2.5) for biogenic and biomass burning emissions and to 0.7 (factor of 2 uncertainty) for the anthropogenic emission fluxes. The non-diagonal elements of

$\mathbf{B}$ are dependent on spatio-temporal correlations of the errors on the fluxes. For biogenic and biomass burning emissions, the spatial correlations are assumed to decrease exponentially with distance $d_{ij}$ between two grid cells $i$ and $j$, such that

$$B_{ij} = \left( \frac{\sigma_i}{\phi_i} \right)^2 \exp\left( \frac{-d_{ij}}{l} \right), \tag{4}$$



where $l$ is the decorrelation length assumed to be 200 km, $\phi_i$ is the total flux emitted from grid cell $i$, and $\sigma_i/\phi_i$ is its relative error. Furthermore, we assume that biogenic emissions emitted by different plant functional types (PFTs) are uncorrelated, such that

$$B_{ij} = \sum_n e_i^n e_j^n \left( \frac{\sigma_i}{\phi_i} \right) \left( \frac{\sigma_j}{\phi_j} \right) \exp \left( \frac{-d_{ij}}{l} \right), \tag{5}$$

with $e_i^n$ being the fraction of flux emitted in cell $i$ by PFT $n$. The spatial correlations of anthropogenic emission errors are assumed constant within country borders to reflect that anthropogenic emission policies are usually taken at country level and that generally uniform methodologies and emission factors are applied within each country in bottom-up inventories. The temporal correlation of the errors is assumed to decrease linearly from 0.7 for consecutive weeks to zero after three months, for all emission categories. Finally, we filter out TROPOMI HCHO columns below $3 \times 10^{15}$ molec. cm$^{-2}$, since these low columns are most uncertain, as discussed in Sect. 4). This filtering of low columns results in too high top-down emissions in northern Europe (see Sect. 5).

The inversion derives updated emission parameters $\mathbf{f}$ that minimize the cost function in Eq. 3. To achieve this, we make use of the adjoint of the MAGRITTEv1.1 model, which calculates the gradient of the cost function with respect to the emission parameters. The cost function is iteratively minimized using a large-scale quasi-Newton method (Gilbert and Lemaréchal, 1989), and a new set of emission parameters $\mathbf{f}$ is calculated. This procedure is repeated until the norm of the gradient of the cost function is decreased by a factor of 100 with respect to its initial value. This is achieved after about 40 to 50 iterations.

## 4   Validation of TROPOMI HCHO over Europe

Previous studies have shown that satellite HCHO data are biased low for high HCHO columns and high for low HCHO columns compared to aircraft (Zhu et al., 2020) and ground-based data (Vigouroux et al., 2020; De Smedt et al., 2021). A careful validation of the satellite product is therefore required in order to prevent the propagation of these biases into the derived top-down emissions. The nature of these biases is unknown at this point. To rule out a possible geographical dependence of the bias, we focus on stations in the European domain to derive the bias correction. We use the FTIR station data described in Sect. 2.2 and perform the validation using daily spatio-temporal collocation criteria, following the approach of Vigouroux et al. (2020). More specifically, we average TROPOMI scenes within 20 km of the station location. Additionally, we select ground-based observations acquired within three hours of the satellite overpass (i.e. from 10:30 to 16:30 LT) in order to minimize the effect of diurnal variations. These criteria provide an optimal balance between data averaging and collocation (see Vigouroux et al., 2020; De Smedt et al., 2021). Days with no valid TROPOMI measurements or no valid ground-based measurement are excluded. For the validation, we only select ground-based and TROPOMI data from May to September, since we use data during this period as constraints for the inversion. Furthermore, we do not consider MAX-DOAS data for TROPOMI validation due to the difference in vertical sensitivities between TROPOMI and MAX-DOAS. The vertical sensitivities of the FTIR and TROPOMI observations have more overlap, specifically in the higher atmospheric layers, whereas the MAX-DOAS





measurement sensitivity is maximum in the boundary layer. FTIR HCHO columns are therefore better suited for validation purposes, whereas MAX-DOAS data are particularly useful to provide complementary information to the satellite columns.

To make a meaningful comparison between space-based and ground-based data, it is necessary to apply a 'smoothing' technique to the ground-based data (Rodgers and Connor, 2003). In this regard, we follow the methodology described in Sect. 4.2 of Vigouroux et al. (2020). First, the vertical profile of HCHO at each station is corrected for the different a priori profile that was used in the retrieval of the satellite column. Using the averaging kernel of the ground-based measurement, the a priori profile is substituted by the TM5-MP profile of the TROPOMI retrieval, thereby modifying the retrieved profile at altitudes where the measurement is not very sensitive. The modified station profile is then smoothed by applying the averaging kernel of the collocated clear-sky TROPOMI measurements. Finally, a correction to the total column is applied to account for topographical elevation variations of the TROPOMI footprint around the station. The smoothed columns can then be directly compared to the satellite columns. In the comparisons, we used weekly averages, since weekly-averaged TROPOMI HCHO columns are used as top-down constraints in the inversion. A comparison between TROPOMI HCHO columns and the smoothed FTIR columns is illustrated in the scatter plot of Fig. 3.

The linear regression is performed using the Passing-Bablok estimator (Passing and Bablok, 1983, 1984; Bablok et al., 1988) which is commonly used in the comparison between measurements of different methods by virtue of its symmetrical properties and its robustness in the presence of outliers. The comparison of the collocated data shows that TROPOMI HCHO columns are biased with respect to FTIR measurements. The FTIR data suggest that the high TROPOMI HCHO columns are too low and the low TROPOMI columns are too high with respect to the FTIR measurements. This agrees well with previous FTIR-based bias corrections over the global domain (Vigouroux et al., 2020). We note that for low TROPOMI HCHO columns, i.e. below $3 \times 10^{15}$ molec. cm$^{-2}$, the regression line falls below a large majority of FTIR data, indicating that it is not suitable in that range.

The statistical uncertainty of the linear fit calculated and shown inset in Fig. 3 is clearly underestimated, given the substantial column differences between the different stations (see also monthly means for individual stations in Fig. S1 of the supplement). We note that stations in northern Europe (Kiruna and Sodankylä) show no significant bias with respect to TROPOMI, while stations at mid-latitudes (Germany and France) show a larger bias. The linear scaling of the bias with respect to the vertical column suggests a large bias in southern Europe. However, due to the lack of ground-based data available in southern Europe, we cannot confirm or infirm this claim at this point. An accurate bias correction is particularly important for southern Europe since the highest biogenic emissions are released from the warmer southern European ecosystems.

An additional source of uncertainty originates in data averaging. Averaging the HCHO data inherently introduces a bias in the fit towards weeks with fewer data. We also find that averaging over longer periods (e.g. monthly) of the satellite data generally results in larger slopes and smaller intercepts in the fit compared to the weekly-based validation. Furthermore, different assumptions in the retrieval of ground-based and satellite HCHO columns might also skew the data set, since negative HCHO columns are absent in the FTIR dataset, whereas they are present in the TROPOMI dataset.

In our inversion scheme, we apply the bias correction based on FTIR data (Fig. 3), given by

$$[\text{TROPOMI}]_{\text{BC}} = 1.48 \times [\text{TROPOMI}] \qquad\qquad - 1.40 \times 10^{15} \text{ molec. cm}^{-2}, \qquad\qquad (6)$$




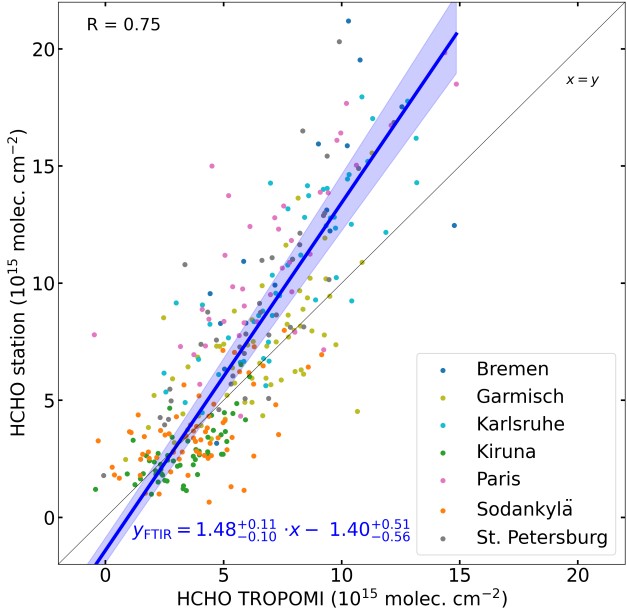

**Figure 3.** Scatter plot of space-based (TROPOMI) vertical HCHO columns versus ground-based (FTIR) vertical HCHO columns. The data have been collocated and weekly averaged. The linear fit is shown in blue, and its formal uncertainties are depicted by the blue-shaded area.

where [TROPOMI] and [TROPOMI]$_{BC}$ are the retrieved and bias-corrected (weekly-averaged) HCHO total columns, respec-

tively. This FTIR-based relation of Eq. 6 is in good agreement with previously published bias corrections of the TROPOMI HCHO product using FTIR (Vigouroux et al., 2020) and MAX-DOAS data (De Smedt et al., 2021). As illustrated in Figs. 4a and 4b, the FTIR-based correction brings small changes to the HCHO levels in northern Europe, but induces substantial increases (30 to 40% over emission sources) in southern Europe.

## 5 Inversion results

### 5.1 Modeled HCHO columns

The a priori and a posteriori model columns are shown in Figs. 4c and 4d, respectively. The a priori column distribution is calculated based on a priori emission inventories as described in Sect. 3. The model computes the HCHO column at the time of the TROPOMI overpass. The modeled vertical profile is then convolved with the averaging kernel of the satellite observation, and a smoothed total column is derived. We note that this implies that model columns are calculated only whenever valid

satellite measurements are available, and the weekly model averages are sampled as the observations. The modeled columns can therefore be directly compared to the TROPOMI HCHO columns of Fig. 4b. The a priori model provides a relatively good agreement with the HCHO observations in terms of spatial distribution with, on average, higher columns in southern Europe ($\sim 10^{16}$ molec. cm$^{-2}$) compared to the north ($\sim 5 \times 10^{15}$ molec. cm$^{-2}$). Some anthropogenic sources appear to be overestimated in



the model, such as the Ruhr area in Germany. Moreover, comparisohttps://www.overleaf.com/project/62d7cd1419ef4c43771b28f2n

of the a priori model with the bias-corrected data shows that the a priori columns in southern Europe are generally too low (see Fig. 4e).

The optimized HCHO columns inferred through inversion of the bias-corrected TROPOMI HCHO columns are shown in Fig. 4d. Because the bias-corrected columns are much higher than in the a priori model in southern Europe, the inversion leads to large column increases of about 2 to $3 \times 10^{15}$ molec. cm$^{-2}$ in the Mediterranean region. This can also be seen in Figs. 4e

and 4f, which illustrate the absolute difference between the bias-corrected observations (Fig. 4b), and either the a priori model or the a posteriori model. The a priori difference map shows clearly the model underestimation in southern Europe. This is largely corrected for by the optimization, which shows a good agreement between model and observations, especially in mainland Europe. In northern Europe and over the seas, differences of the order of $1 \times 10^{15}$ molec. cm$^{-2}$ are seen. Even higher discrepancies are found over the northern seas ($> 65°$ N) which is caused by the large uncertainties of the satellite product and

by the filtering out of low weekly columns ($< 3 \times 10^{15}$ molec. cm$^{-2}$). This filter leads to a positive bias of the temporally-averaged columns in regions where the columns are frequently lower than this threshold, as is the case over northern seas and Scandinavia. Furthermore, whereas the precision of weekly-averaged TROPOMI columns amounts to only a few percent of the column and plays a negligible role in southern Europe, the high cloudiness and low columns found at high latitudes lead to larger random errors, sometimes of the same order as the columns, thereby aggravating the potential bias due to the

threshold. The inversion results in these regions should therefore be considered with caution. We also note that oceanic data are excluded from the inversion, because the columns over oceans are mostly due to the oxidation of methane (Stavrakou et al., 2009). Consequently, the comparison between model and observations does not change significantly over the northern seas. We note however a substantial reduction of the bias over the Mediterranean and Black seas, due to emission changes over land and subsequent transport of HCHO precursors (or of intermediates in their oxidation mechanisms) to those areas.

Figure 5 shows time series of observed and modeled HCHO columns in 2019 for four regions in the European domain: Germany, central Europe, Iberia, and Anatolia. The optimized model based on our standard inversion described in Sect. 3 uses weekly TROPOMI observations (orange curve). Figure 5 includes the results of an inversion using monthly TROPOMI HCHO columns in order to evaluate the effect of temporal averaging on the top-down emissions (see also scenario 10 in Sect. 6 and Table 3). The emission increments of the monthly inversion run are applied to the weekly a priori HCHO columns in order to

allow for comparison with the standard weekly inversion results.

Over Germany, the TROPOMI HCHO columns have a larger variability than the a priori model. The weekly optimization improves the model by increasing the HCHO columns during peaks and decreasing the model during troughs in the observed time series. The monthly inversion does not affect the weekly variability of the model, therefore the Pearson correlation coefficient of the monthly optimization ($R = 0.84$) is lower than the standard weekly optimization ($R = 0.92$). The second region

of Fig. 5 focuses on central Europe, more precisely the Pannonian Basin. This region was chosen for its biogeographical and climatological uniformity, as it is surrounded on all sides by mountain ranges. Similar to Germany, we find a large improvement in the correlation coefficient for the weekly top-down inversion (from 0.92 to 0.98), where the high and low HCHO columns in the time series are much better reproduced. For regions Iberia and Anatolia, both in southern Europe, the results of





the weekly and monthly optimizations are similar. The TROPOMI HCHO columns are much higher than the a priori model,
which is partly due to the effect of the bias correction. The top-down HCHO columns follow the high observed columns as
expected. Due to the relatively constant meteorology in southern Europe (i.e., high temperatures and few clouds), the variabil-
ity of HCHO columns is much lower. Consequently, the difference between the weekly and monthly inversion run is relatively
small.

## 5.2    Top-down emissions

Figure 6a shows the a priori emissions of the MEGAN-MOHYCAN inventory, which were used as input for the top-down
inversion. High-emission regions generally have a warm climate, along with vegetation consisting of strong isoprene emitters.
The a priori emissions are heavily dependent on assumptions regarding the variable and uncertain emission factors of the
different vegetation types, as well as on the land-use map that enters the emission model. Figure 6b shows top-down isoprene
emissions optimized based on bias-corrected TROPOMI HCHO columns. The emission enhancement is presented in Fig. 6c,
calculated as the ratio of the top-down emissions over the a priori emissions. The increment map shows increases in the
emissions across the majority of the domain, except for some regions between $50°$ and $60°$ N. Especially in southern Europe,
the data suggest higher isoprene emissions in order to explain the high observed TROPOMI HCHO columns. Similarly, at high
latitudes there are increased top-down isoprene emissions, but this is likely due to the positive bias in this region introduced by
the filtering of low TROPOMI HCHO columns.

Table 2 presents the annual emissions in the European domain. The inversion strongly increases isoprene emissions, by more
than a factor of two with respect to the a priori (from 8.1 Tg yr$^{-1}$ in MEGAN-MOHYCAN to 18.5 Tg yr$^{-1}$ in the top-down
emissions). In a previous inversion study by Bauwens et al. (2016) using OMI data, only a mild increase of isoprene emissions
was derived over Europe. This is mainly due to the absence of a bias correction in their inversion study. The results of Bauwens
et al. (2016) suggested top-down isoprene emissions across the domain of 8.4 Tg yr$^{-1}$ over the years 2005 to 2013, i.e. 23%
higher than their a priori biogenic emissions. The majority of the biogenic emissions take place in southern Europe, where the
high observed HCHO columns lead to a strong increase of the top-down emissions in our inversion. The time series of isoprene
emissions in Europe (Fig. 7a) shows that the biogenic emission increases are mainly occurring during the summer peak from
June to August with up to a factor of three higher emissions. As expected, the emissions are close to zero during winter. We note
that the emissions are optimized using TROPOMI observations only during summer months, but due to temporal correlations
imposed in the inversion, the emissions during winter months are also affected.

The biomass burning VOC emissions of the top-down model are on average higher than the QFED a priori inventory. The
most prominent source of biomass burning VOCs in Europe is due to agricultural waste burning. This is prominent in the time
series in Fig. 7b, which shows peaks around April and September mainly due to agricultural fires in Ukraine and Russia (see
Fig. S3). Figure 7b also shows substantial year-to-year variability, in which the top-down inversion does not always lead to
changes in the emissions with respect to the QFED a priori inventory. However, the data on average lead to higher top-down
fire emissions which are mainly caused by increased agricultural fire VOC emissions in eastern Europe and the Danube river
valley, as can be seen in Fig. S3c.





In terms of anthropogenic emissions, the optimized fluxes are lower than the a priori fluxes from CAMS-GLOB-ANT by about 20%. This decrease is driven by generally lower observed HCHO columns over cities and industrial regions. Furthermore,

the TROPOMI HCHO columns are lower than that of the a priori model during colder weeks (see Fig. 5), which suggests a larger relative contribution of biogenic emissions to the total column. Figure S4 shows that the optimized emissions are lower than CAMS-GLOB-ANT for most countries at mid-latitudes (between roughly 45° and 55° N), with the largest differences occurring in Germany. The effect of the imposed country-dependent spatial correlations of the emissions parameters is clearly visible, as the emission increments of anthropogenic VOCs are similar within country borders. The time series in Fig. 7c

depicts a weak seasonal cycle with higher emissions in winter compared to summer. This seasonal cycle is preserved in the top-down fluxes, but with lower values during the whole year.

## 5.3 Model evaluation

We evaluate the model against ground-based remote sensing and in situ measurements. For the FTIR and MAX-DOAS comparisons, we select model data at the time of the measurement at the $0.5° \times 0.5°$-pixel in which the station is located. The model

profile is smoothed using the averaging kernel of the measurement. Comparison of the optimized model to FTIR station data shows an improvement with respect to the a priori model at Bremen, Garmisch, Karlsruhe, and Paris, as shown by the reduction of average absolute deviations (AAD) at those stations (see Fig. 8). The opposite effect occurs at the northern stations (Kiruna, Sodankylä, and St. Petersburg), where the inversion worsens the agreement between model and data. More specifically, model overestimations are found at high-latitude Scandinavian sites, invalidating the emission increases in northern Scandinavia. At

St. Petersburg, the TROPOMI HCHO columns are generally very low, leading to a decrease in the modeled HCHO columns after the inversion, yet FTIR data suggest higher HCHO columns. The reduction of emissions that follows from the inversion in this region (around 60° N, see Fig. 6) is therefore not corroborated by the FTIR data at St. Petersburg.

The comparison of the model columns with the MAX-DOAS measurements of stations listed in Table 1 is shown in Fig. 9. For all stations (except Athens), the optimized HCHO columns are not improved with respect to the a priori, and the average

absolute difference is increased at several locations. The MAX-DOAS data show a worsening of the optimized model over Germany and the Benelux, whereas the FTIR data suggest an improvement of the model over Germany and France. This discrepancy is potentially linked to FTIR needing clear-sky conditions for measurements, whereas MAX-DOAS also observes during cloudy days. The optimized emissions have increased biogenic emissions over Germany and the Benelux (see Fig. 6c), but decreased anthropogenic VOC emissions in this region (Fig. S4). This leads to lower HCHO columns during colder, cloudy

days, which is also visible in the HCHO time series over Germany in Fig. 5. These emission increments are at odds with the MAX-DOAS data, which suggest larger HCHO columns, specifically during cloudy weeks. However, the presence of clouds could potentially lead to biases in the MAX-DOAS HCHO columns. In southern Europe, the higher emissions of the optimized model are corroborated by the Athens MAX-DOAS data. This station is located at an altitude of 500 m, which is higher than the city itself. Since MAX-DOAS observations only consider upward looking elevation angles, an important part of the HCHO

column is not measured. We take this into account by only including part of the model column at 500 m altitude and higher for





the comparison. The increased biogenic VOC emissions around Athens lead to increased HCHO mixing ratios of the model above 500 m altitude, which agrees with the MAX-DOAS data at Athens.

As an additional evaluation, we compare the modeled near-surface HCHO concentrations to in situ HCHO data in Fig. 10 and Table S1. We sample the model at the pixel and vertical layer corresponding to the coordinates and altitude (above sea level)
of the station and compute the ratio of monthly-averaged daytime (usually 8–12 h) modeled concentration to the corresponding monthly-averaged collocated in situ measurements. The ratio is computed using a climatological average if all observations are dated before May 2018 (see Table 1). We find that for the two Spanish stations (San Pablo de los Montes and Campisábalos), the a priori model surface concentration is already higher than the measurements. This discrepancy is increased even further after the optimization. As for the other southern European stations, we find improvements in the south of France (La Tardière
and Peyrusse Vieille), and minor improvements for the Italian stations (Ispra and Montelibretti) where the a priori model is about 25% too low, but the optimized model is 10 to 20% higher than the observations. For the stations located at higher latitudes, we generally see a small effect of the inversion on the optimized model, with the a priori performing better for the Waldhof and Brotjacklriegel stations. We note that some caution is needed for comparisons that are not temporally collocated (all except LAH, PAB, PEY, and TAR, see Table 1), since meteorology has an important effect on HCHO concentrations, and
for the stations located in mountainous terrain (PAB, CAM, BRO, DON, and SCH), due to local orography-induced transport patterns impacting the concentrations.

To summarize, the optimized model provides better agreement with the FTIR stations below about 55° N and with the only southern MAX-DOAS station Athens. The other MAX-DOAS stations located in Germany and the Benelux display higher HCHO columns than the a priori model, yet the (bias-corrected) TROPOMI HCHO columns do not suggest an increase of the
emissions in this area. The three FTIR stations in northern Europe compare better with the a priori than with the optimized model, thereby contradicting the emission increases in northern Scandinavia and the emission decreases in the St. Petersburg area. In southern Europe, we see large increases for biogenic emissions of the optimized model. This is corroborated by HCHO column measurements at the Athens station, although in situ measurements provide mixed results in Spain, France, and Italy.

## 6 Sensitivity inversions

The results of the top-down inversion are subject to uncertainties related to the model, the inversion setup, the a priori inventories, and the observations. These uncertainties are difficult to quantify. In this section, we present a range of ten sensitivity inversions aimed at evaluating the effect of different sources of uncertainty on the derived top-down emissions. We provide an overview of the different sensitivity runs (S1 to S10) in Table 3. Each sensitivity run was performed for the year 2019 and their annual emissions are shown in Fig. 11.
The first scenario (S1) described in Table 3 is an inversion constrained by TROPOMI HCHO columns without bias correction. The data used to constrain the inversion in this scenario is depicted in Fig. 4a, as opposed to Fig. 4b for the standard inversion scenario (S0). Because the bias correction leads to much higher HCHO columns over strongly emitting regions (~ 40% increase), the S1 run has much lower emissions across Europe for all emission categories. Since the effect of the bias





correction is strongest in southern Europe, the biogenic emissions are most affected, with the S1 scenario having three times

lower isoprene emissions compared to the standard run (6.7 Tg yr$^{-1}$ versus 18.3 Tg yr$^{-1}$ for S0). The S1 results highlight the strong impact of TROPOMI observations on the top-down emissions, and therefore the importance of a comprehensive and homogeneous validation dataset.

Next, we evaluate the effect of different a priori inventories on the inversion results. In scenario S2, we substitute MEGAN-MOHYCAN biogenic emissions with the CAMS-GLOB-BIOv3.1 inventory (Sindelarova et al., 2022). Figure 12a shows the

isoprene emissions of CAMS-GLOB-BIOv3.1 gridded to $0.5° \times 0.5°$ and averaged from May to September in 2019. On average, the CAMS-GLOB-BIOv3.1 isoprene emissions are much lower than MEGAN-MOHYCAN, with annual total emissions of 4.1 Tg for CAMS-GLOB-BIOv3.1 versus 7.9 Tg for MEGAN-MOHYCAN. Furthermore, the distribution of emissions is very different, especially in Turkey, which is a major source region of biogenic emissions in MEGAN-MOHYCAN. Those differences in a priori emissions have a major effect on the optimized top-down emissions, which are shown in Fig. 12b. The

optimized total isoprene emissions are more than 30% lower in the S2 run, while the effect on the optimized fire and anthropogenic VOC emissions is limited. Although the top-down biogenic emissions of the S2 run are lower than in the standard S0 run, we still see a large increase with respect to the a priori emissions of about 300%. Similar to the standard case, the emissions are mainly increased in southern Europe, and most heavily in Turkey by over a factor of ten (see Fig. 12c).

In scenario S3, the a priori biomass burning emissions are substituted with the GFED4s inventory (van der Werf et al., 2017).

The GFED4s VOC emissions are based on a burned area approach and are on average lower than the QFED2.4 emissions used in our standard inversion, based on a fire radiative power approach (Darmenov and da Silva, 2015). However, there is a good agreement between the two inventories in terms of temporal and spatial occurrence of fire events, as they both make use of MODIS satellite data (Pan et al., 2020). The lower GFED4s emissions of scenario S3 result in lower top-down fire emissions, totalling 2.1 Tg VOC for the S3 run as opposed to 3.5 Tg VOC for the standard S0 run. Since biomass burning is not a dominant

emission source of VOCs in Europe, the effect of differences in a priori emissions has only a limited impact on the top-down biogenic and anthropogenic VOC emissions.

The S4 scenario tests an inversion using biomass burning emission factors from Akagi et al. (2011), instead of emission factors from Andreae (2019) used in the standard inversion. Using the Akagi et al. (2011) emission factors, the total emission of pyrogenic VOCs is decreased by about 20% in the a priori and optimized emissions. This can be mainly attributed to

differences in emissions for agricultural waste burning. Specifically, the emission factors of large alkanes (butane or higher) is substantially lower in the S4 run. As for the S3 scenario, the lower a priori emissions lead to lower a posteriori pyrogenic emissions. The impact of this sensitivity run on the optimized biogenic and anthropogenic emissions is small.

Model assumptions represent an important source of uncertainty in the inversions. In the next scenarios, we evaluate the effects of some of these assumptions on the top-down results. The S5 and S6 runs evaluate the effect of the assumed errors

on the emission parameters of the a priori inventories by decreasing and increasing the errors, respectively. As described in Sect. 3, because bottom-up inventories have no quantified uncertainties, we assume relative errors on the flux parameters corresponding to a factor of 2.5 for biogenic and biomass burning emissions, and a factor of 2 for anthropogenic emissions. By decreasing (increasing) these errors, the inversion will be less (more) inclined to depart from the a priori in order to match





the observations. Indeed, the S6 run, in which the errors are increased by a factor of 1.5, differs more strongly from the a priori
inventories compared to the standard S0 run. Similarly, the S5 run with 1.5 times lower errors leads to smaller increases of
biogenic emissions and smaller decreases of anthropogenic emissions. The effect of changing the error parameters is strongest
for the anthropogenic emissions ($\sim 20\%$), though overall the effect of changing the errors is limited (less than 10% for biogenic
emissions).

Additionally, the correlations between errors on biogenic and pyrogenic emissions are assumed to decay exponentially with
distance between the grid cells, for which the decorrelation length is taken to be 200 km (see Eqs. 4 and 5). The effect of the
decorrelation length is equivalent to smoothing the emissions by introducing a spatial correlation of the emission parameters.
In sensitivity inversion scenarios S7 and S8, we test the effect of a shorter (100 km) and longer (400 km) decorrelation length
on the inversion results. Increasing the decorrelation length leads to overall increased biogenic emissions, since the increased
emission parameters in southern Europe are forced over longer distances. Conversely, the S7 run with half the decorrelation
length leads to about 10% lower biogenic emissions. As expected, the anthropogenic emissions are less affected by these
changes.

Besides the inversion setup, the chemical mechanism is an additional source of uncertainty. In particular, the oxidation of
isoprene by OH radicals is a complex process, and the initial oxidation steps are still an active research topic (Müller et al.,
2019; Møller et al., 2019; Novelli et al., 2020; Schwantes et al., 2020; Medeiros et al., 2022). For example, the isomerization
rates of specific isoprene peroxy radicals remain uncertain. In the standard run, we apply 1,6 H-shift rates based on the LIM1
mechanism (Peeters et al., 2014) in agreement with a recent laboratory determination Novelli et al. (2020). These rates are
higher than those given in Müller et al. (2019), which were derived by Wennberg et al. (2018) based on experimental work at
Caltech (Teng et al., 2017). The lower isomerization rate leads to a higher HCHO yield from the oxidation of isoprene, which
therefore leads to higher HCHO columns. The reason lies in the lower HCHO yields from the further oxidation of isomerization
products like the hydroperoxy aldehydes HPALDs (Peeters et al., 2014; Müller et al., 2019). The optimization of this scenario
(S9) results in total isoprene emissions of 17.2 Tg yr$^{-1}$, i.e. 6% lower than in the standard run (18.3 Tg yr$^{-1}$). This reduction is
largely compensated by higher pyrogenic and anthropogenic emissions, such that the total VOC emission of S9 is only slightly
(2%) lower than in S0.

In all previous scenarios (S0 to S9), weekly-averaged TROPOMI HCHO columns were used to constrain the emissions,
and we update the emissions at weekly increments. In the S10 run, we evaluate the impact of temporal averaging by using
monthly-averaged HCHO columns and monthly emission parameters in the inversion. In Sect. 5.1, we showed that the monthly
inversion is less capable to match the weekly variability in the HCHO time series (see Fig. 5). Furthermore, the emissions of
the S10 run are closer to the a priori. For example, the annual anthropogenic VOC emissions of the S10 run are less decreased
($-12\%$) with respect to the a priori compared to the weekly S0 run ($-21\%$), whereas the biogenic emission increase shows
the opposite effect. This can be expected since the monthly inversion has about four times less observational constraints than
the weekly inversion, which results in smaller departures from the a priori. The number of emission parameters is also lower
in the monthly inversion, but this effect is mitigated due to temporal correlations between emission parameters. Additionally,
the monthly inversion is less capable of disentangling the biogenic and anthropogenic emissions sources, since it averages





out meteorological variability at weekly time scales, which is a major driver of biogenic emission variability. The Benelux region provides an interesting example, since the weekly inversion displays increased biogenic emissions whereas the monthly inversion leads to decreased biogenic emissions. The strong variability of the HCHO columns in the Benelux is displayed in Fig. 2. In order to explain the large fluctuations in the observations, biogenic emissions are increased by the inversion during warm weeks, and anthropogenic emissions are decreased in order to account for the low columns during cold weeks.

Over northern Europe, the results of the monthly inversion are very different from the standard weekly inversion. In particular, the S10 optimization does not display the increase in the emissions above $60°$ N (see Figs. S5 and S6 of the supplement). As previously discussed, the increased biogenic emissions in Scandinavia in the standard run is due to the filtering out of low HCHO columns ($< 3 \times 10^{15}$ molec. cm$^{-2}$) implying that only high columns (i.e. positive outliers) remain in the data set. However, the monthly averaging of the S10 run reduces the noise in the TROPOMI HCHO columns, leading to less filtering of low values and therefore to lower average HCHO columns at high latitudes. Consequently, the monthly inversion does not show an increase of emissions in Scandinavia and provides more reliable results in this region. In conclusion, the weekly inversion provides generally improved results compared to the monthly inversion, except at northern latitudes (above about $60°$ N).

To summarize, by far the largest uncertainty in our top-down emissions arises from the application of the bias correction. This emphasizes the importance of continuous and high-quality validation of the HCHO satellite product. This is especially important in southern Europe, where the bias correction has the biggest impact and where validation data are scarce. The biogenic a priori inventory also plays an important role in the inversion, mainly due to the large differences among the available bottom-up inventories. The range of different inversion settings and model parameters that we have tested results in typical differences of roughly 10 to 20% on the final total emission fluxes, as can be seen in Fig. 11.

## 7    Conclusions

We have performed a comprehensive top-down inversion of VOC emissions over Europe from 2018 to 2021 constrained by TROPOMI HCHO columns. We have used the MAGRITTEv1.1 CTM and its adjoint to optimize biogenic, pyrogenic, and anthropogenic VOC emissions at a weekly temporal resolution. The inversion suggests a strong underestimation of MEGAN emissions, especially in southern Europe, with the total annual isoprene flux increasing from 8.1 Tg yr$^{-1}$ to 18.5 Tg yr$^{-1}$. This large increase is driven by the high (bias-corrected) HCHO columns observed by TROPOMI during the summer months in southern Europe. The biomass burning VOC emissions show a slight increase (from 2.3 to 2.6 Tg yr$^{-1}$) in the inversion, with a strong interannual variability of the emission increments. The anthropogenic VOC emissions are decreased by the inversion (from 16.9 to 14.0 Tg yr$^{-1}$), with the main reduction occurring over Germany.

Comparison of the TROPOMI HCHO columns and ground-based FTIR HCHO columns shows that TROPOMI HCHO columns are generally too low, especially for high columns. We derived a linear relationship characterizing the bias against FTIR data at European stations (excluding mountain sites), which agrees with the bias correction reported in previous validation studies using global FTIR and MAX-DOAS networks (Vigouroux et al., 2020; De Smedt et al., 2021). Additionally, we used FTIR, MAX-DOAS, and in situ data to validate the top-down model results and found that the optimized model performs well



for FTIR stations below about 55° N. In northern Europe, where the signal-to-noise ratio of TROPOMI data is usually low, our top-down emissions are too high due to the filtering of low HCHO columns in the inversion. At MAX-DOAS stations, the optimized model shows no improvement, or even a worsening of the comparison with ground-based measurements, except at Athens. The MAX-DOAS data suggest that the emissions are too low in the Benelux and Germany, specifically during colder and cloudy days. Finally, the comparison with in situ data shows improved results at several locations (south of France, Italy) but too high concentrations of the model at other places, including the Spanish stations. These mixed results might reflect the limited representativeness of point measurements and/or difficulties with model, e.g. regarding boundary layer mixing.

To evaluate the robustness of the inversion results, we have performed a range of sensitivity inversions. As expected, the top-down emissions are very sensitive to the observed HCHO columns, and therefore also to the bias correction of the TROPOMI data. This highlights the importance of comprehensive and extended validation studies, using network remote-sensing data and/or airborne validation campaigns. We note a scarcity of validation data in southern Europe, although at the time of writing, there are plans for FTIR measurement sites in Italy and Cyprus. Additionally, we find that the choice of a priori biogenic emissions plays an important role due to large differences between available bottom-up emission inventories. Other model-related effects on the inversion induce differences of up to 10% on the final annual total emissions.

Finally, the inversion using weekly averages was found to provide better results than the monthly inversion run, except in the northernmost part of the continent. Especially regions with variable meteorology are better treated by the weekly inversion due to the strong fluctuations in the HCHO columns. As the HCHO data quality continues to improve with the advent of more advanced satellite missions, the observational constraints on the inversion become more tight allowing for higher spatio-temporal resolution of the inversion studies. In this European inversion study, we found that weekly averages and a spatial resolution of 0.5° are convenient in terms of signal-to-noise ratio and data availability. However, it may be possible to perform higher resolution inversions in other regions in the world characterized by strong emissions (hence a strong HCHO signal) and moderate or low cloudiness.

*Data availability.* The TROPOMI HCHO dataset is a Copernicus operational product and is available at https://doi.org/10.5270/S5P-tjlxfd2. The FTIR and MAX-DOAS datasets can be requested by contacting the PIs of each station. In situ data from the EMEP network is available at the EBAS data portal (https://ebas.nilu.no). The top-down biogenic and biomass burning emission datasets generated in this study are available at SEEDS project data portal (https://www.seedsproject.eu/data) and at the BIRA-IASB emission portal (https://emissions.aeronomie.be). The MEGAN-MOHYCAN bottom-up inventory is also available at https://emissions.aeronomie.be.

*Author contributions.* GMO carried out the analysis and wrote the manuscript. JFM and TS designed the inversion scheme and performed the optimizations. IDS described the TROPOMI HCHO data in Sect. 2.1. TB, RK, MM, MP, ARo, YT and CV provided the FTIR measurements. CV provided the description of the FTIR datasets in Sect. 2.2. MMF, UF, FH, AM, AP, ARi, MVR, and TW provided the FRM$_4$DOAS measurements. AM provided the description of the FRM$_4$DOAS data (Sect. 2.3). All authors read and commented on the manuscript.



.

*Competing interests.* Andreas Richter, Thomas Wagner, and Michel Van Roozendael are editors of the journal.

*Acknowledgements.* This research was performed as part of the SEEDS project funded by the European Commission under the H2020 programme (grant agreement no. 101004318, 2021–2023). The FTIR data was made available as part of the PRODEX TROVA-E2 Belspo project (2021–2023). The Paris TCCON site has received funding from Sorbonne Université, the French research center CNRS, the French space agency CNES, and Région île-de-France. We acknowledge the $FRM_4DOAS$(-2.0) project under ESA contract numbers 4000118181/16/I-EF and 4000135355/21/I-DT-Ir.



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





**Table 1.** List of stations providing ground-based remote sensing measurements (FTIR and MAX-DOAS) and in situ measurements.

FTIR stations

| Station | Code | Latitude (° N) | Longitude (° E) | Altitude (m) | Years | # days |
|---|---|---|---|---|---|---|
| Bremen | BRE | 53.10 | 8.85 | 30 | 2018–2020 | 23 |
| Garmisch | GAR | 47.48 | 11.06 | 743 | 2018–2021 | 141 |
| Karlsruhe | KAR | 49.10 | 8.42 | 110 | 2018–2020 | 136 |
| Kiruna | KIR | 67.84 | 20.40 | 420 | 2018–2021 | 116 |
| Paris | PAI | 48.97 | 2.37 | 60 | 2018–2021 | 94 |
| Sodankylä | SOD | 67.37 | 26.63 | 190 | 2018–2021 | 203 |
| St. Petersburg | SPE | 59.88 | 29.83 | 20 | 2018–2021 | 94 |

MAX-DOAS stations

| Station | Code | Latitude (° N) | Longitude (° E) | Altitude (m) | Years | # days |
|---|---|---|---|---|---|---|
| Athens | ATH | 38.05 | 23.86 | 532 | 2018–2021 | 388 |
| Bremen | BRE | 53.10 | 8.85 | 46 | 2018–2021 | 301 |
| Cabauw | CAB | 51.97 | 4.93 | 3 | 2021 | 67 |
| DeBilt | DEB | 52.10 | 5.18 | 22 | 2020–2021 | 87 |
| Heidelberg | HEI | 49.42 | 8.67 | 145 | 2018–2021 | 314 |
| Mainz | MAZ | 49.99 | 8.23 | 150 | 2018–2021 | 220 |
| Uccle | UCC | 50.80 | 4.36 | 95 | 2018–2021 | 139 |

In situ stations

| Station | Code | Latitude (° N) | Longitude (° E) | Altitude (m) | Years | # days |
|---|---|---|---|---|---|---|
| Lahemaa | LAH | 59.50 | 25.9 | 32 | 2018–2019 | 201 |
| Peyrusse Vieille | PEY | 43.62 | 0.18 | 200 | 2018–2020 | 280 |
| San Pablo de los Montes | PAB | 39.55 | -4.35 | 917 | 2018–2021 | 209 |
| La Tardière | TAR | 46.65 | -0.75 | 133 | 2018 | 55 |
| Birkenes | BIR | 58.38 | 8.25 | 190 | 1994–2005 | 613 |
| Brotjacklriegel | BRO | 48.82 | 13.22 | 1016 | 1999–2004 | 477 |
| Campisábalos | CAM | 41.27 | -3.14 | 1360 | 2004–2009 | 551 |
| Donon | DON | 48.50 | 7.13 | 775 | 1994–1999 | 481 |
| Ispra | ISP | 45.80 | 8.63 | 209 | 1994–1997 | 288 |
| Košetice | KOS | 49.57 | 15.08 | 535 | 1994–2005 | 1097 |
| Montelibretti | MON | 42.10 | 12.63 | 48 | 1994–1996 | 122 |
| Schmücke | SCH | 50.65 | 10.77 | 937 | 1999–2004 | 516 |
| Waldhof | WAL | 52.80 | 10.76 | 74 | 1999–2004 | 535 |
| Zingst | ZIN | 54.44 | 12.72 | 1 | 1999–2004 | 533 |



**Figure 4.** HCHO columns over the European domain averaged from May to September for years 2018 to 2021. Panels (**a**) and (**b**) in the top row show TROPOMI HCHO columns without bias correction and with the FTIR-based bias correction from Eq. 6, respectively. The latter is used as constraints in the inversion framework. The middle row contains modeled HCHO columns computed by the MAGRITTEv1.1 model. Panel (**c**) depicts the a priori HCHO columns, and panel (**d**) shows the columns based on the optimized emissions resulting from the top-down inversion. Finally, panel (**e**) is the absolute difference between the bias-corrected TROPOMI observations (panel **b**) and the a priori model (panel **c**), and panel (**f**) shows the absolute difference between panel (**b**) and the optimized model (panel **d**).



**Figure 5.** (**a**) Regions used for averaging in the following panels. (**b**)–(**e**) Weekly time series of HCHO columns of TROPOMI observations (black), MAGRITTE a priori (blue), top-down weekly inversion (orange), and top-down monthly inversion (purple). The HCHO columns cover May to September for the year 2019. In each panel, the Pearson correlation coefficient is indicated for the a priori model (blue), and weekly (orange) and monthly (purple) inversion results.



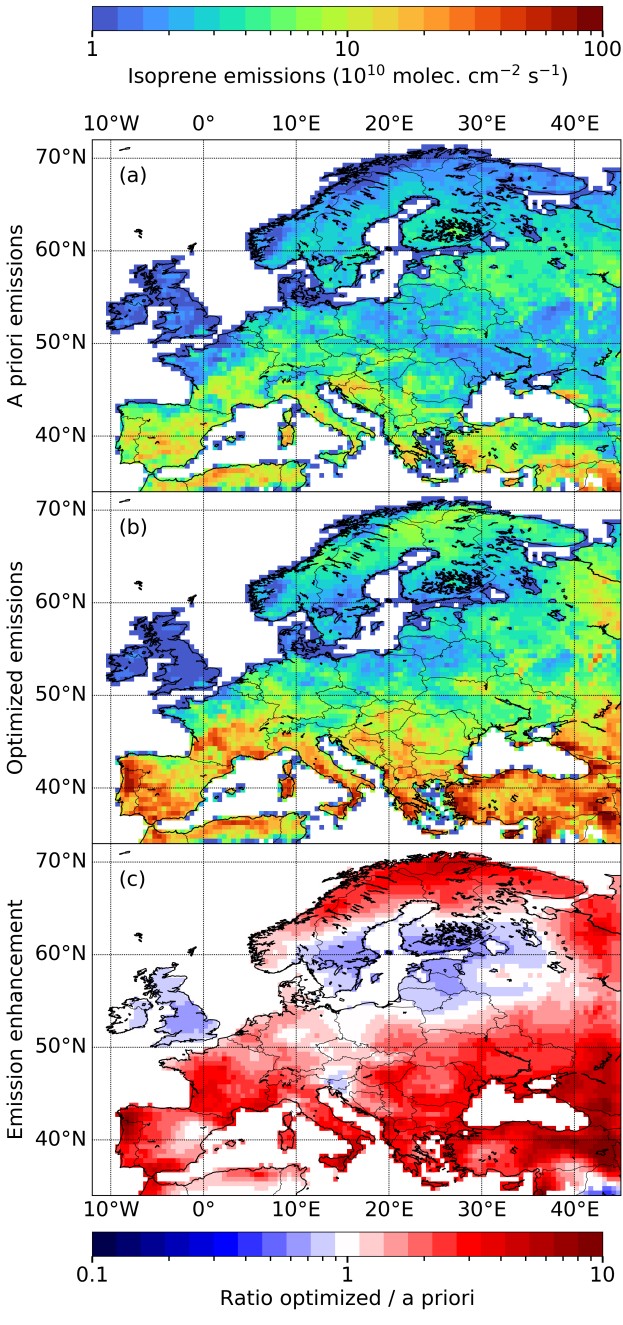

**Figure 6.** Results of the inversion of isoprene emissions over Europe. The top panel (**a**) shows isoprene emissions from the MEGAN-MOHYCAN inventory used as input for the inversion. The middle panel (**b**) shows the optimized emissions resulting from our top-down inversion. The bottom panel (**c**) shows the enhancement map derived as the ratio of the optimized and the a priori emissions (**b** divided by **a**). The data are averaged over the summer months (May to September) from 2018 to 2021.





**Figure 7.** Interannual time series of monthly biogenic emissions (**a**), biomass burning VOC emissions (**b**), and anthropogenic VOC emissions (**c**) across the European domain in Tg. The a priori values are denoted by the solid lines and the optimized values shown by the dotted lines.

**Table 2.** Average annual emissions in $\mathrm{Tg\,yr^{-1}}$ of the three categories of VOC emissions in the a priori and optimized models for the years 2018 to 2021.

| Emission category | a priori | optimized |
|---|---|---|
| Isoprene | 8.1 | 18.5 |
| Biomass burning VOCs | 2.3 | 2.6 |
| Anthropogenic VOCs | 16.9 | 14.0 |







**Figure 8.** Time series of monthly averages of observed FTIR and a priori and a posteriori modeled HCHO columns from May to September. The modeled columns are calculated using either a priori or optimized emissions. The model columns are smoothed using the averaging kernels of the collocated measurements. The average absolute deviation (AAD) between the observations and model, calculated as the mean of $|\mathrm{HCHO_{obs}} - \mathrm{HCHO_{model}}|$ in units of $10^{15}\ \mathrm{molec.\,cm^{-2}}$, is given at the top right of each panel.







**Figure 9.** Same as Fig. 8, but with MAX-DOAS HCHO columns.



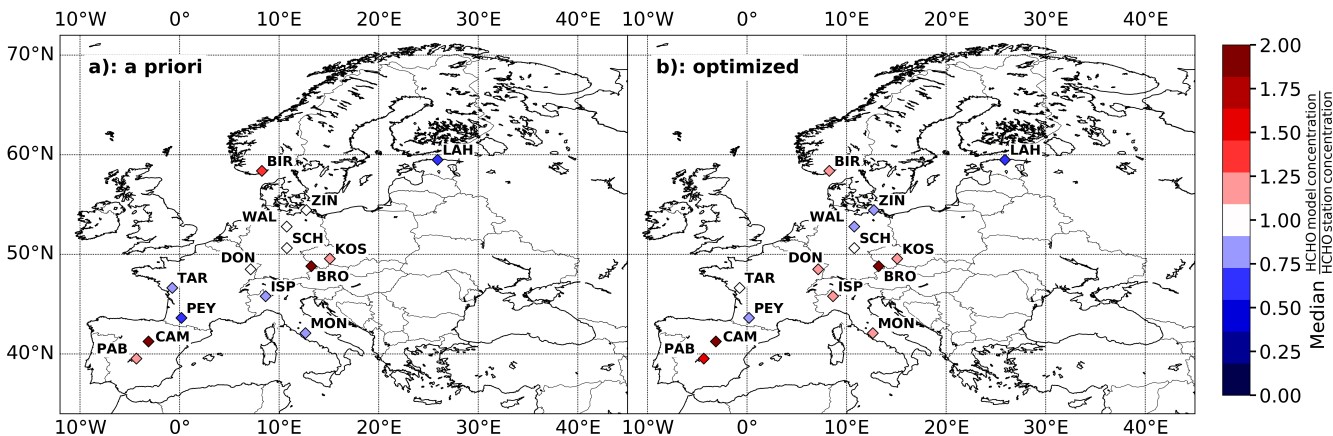

**Figure 10.** Comparison between in situ measurements and near-surface concentrations from the a priori simulation (panel **a**) and from the optimized model (panel **b**). The data are monthly averaged and only data from May to September are included in the ratio calculation. The color of the station symbol represents the median ratio of the model concentration and the station concentration. The ratios calculated with the a priori and optimized model are given in Table S1 of the supplement. Stations with data after 2018 use collocated model concentrations for the comparison. For other stations with only older data, climatological averages are used (see Table 1).



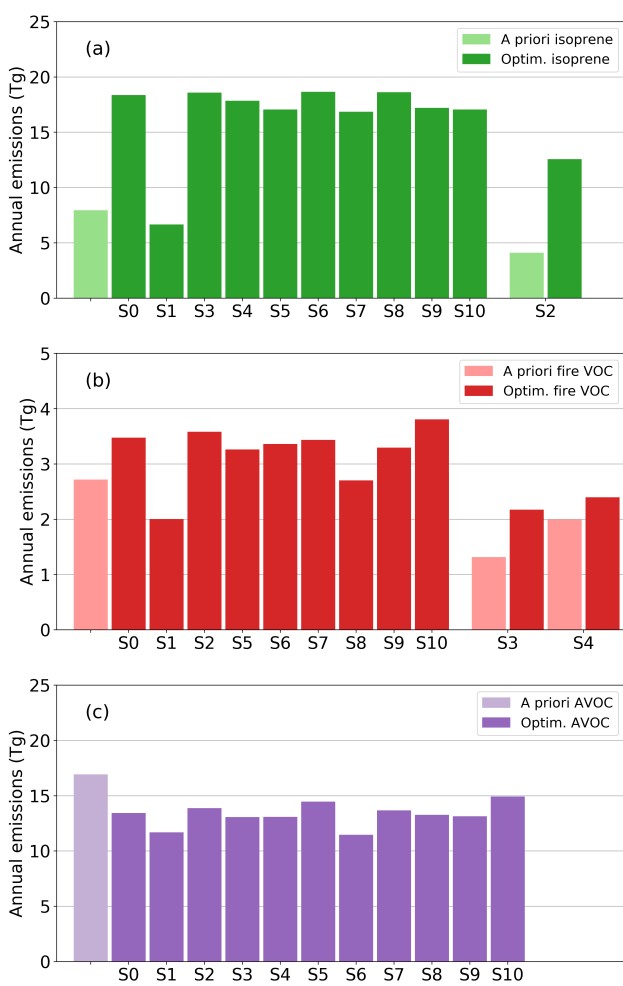

**Figure 11.** Bar chart of annual emissions of the a priori, the standard inversion run (S0), and the sensitivity inversions (S1 to S10) for biogenic emissions (panel **a**), biomass burning emissions (panel **b**), and anthropogenic VOC emissions (panel **c**). The emissions are calculated for the year 2019.



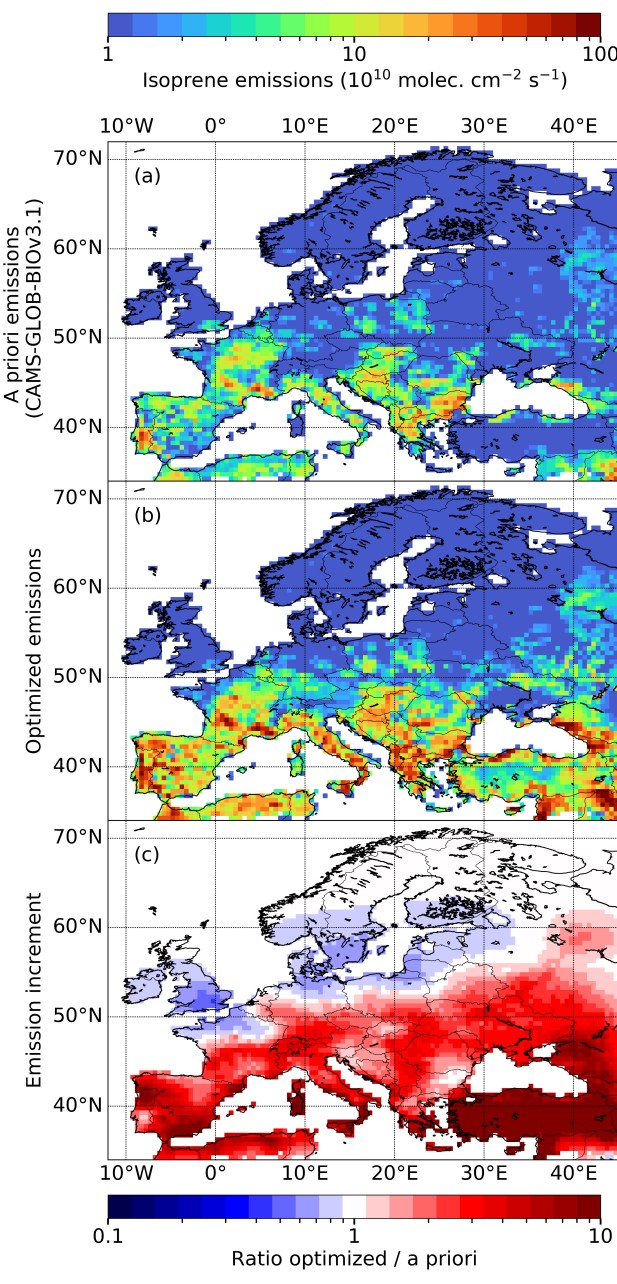

**Figure 12.** Same as Fig. 6, but for scenario S2 using the CAMS-GLOB-BIOv3.1 inventory as a priori biogenic emissions (panel **a**). Panel (**b**) shows the optimized emissions of scenario S2, and panel (**c**) shows the ratio of the optimized and the a priori emissions (**b** divided by **a**). The data are averaged over the summer months (May to September) in 2019.



**Table 3.** Sensitivity inversions conducted in this study, and annual VOC emissions in Tg yr$^{-1}$ over the model domain. AVOC stands for anthropogenic VOC.

| Scenario | Description | Isoprene (Tg) | | Fire VOC (Tg) | | AVOC (Tg) | |
|---|---|---|---|---|---|---|---|
| | | a priori | optim. | a priori | optim. | a priori | optim. |
| S0 | Standard inversion run described in Sect. 3 | 7.9 | 18.3 | 2.7 | 3.5 | 16.9 | 13.4 |
| S1 | as S0, but using TROPOMI HCHO columns without bias correction | 7.9 | 6.7 | 2.7 | 2.0 | 16.9 | 11.7 |
| S2 | as S0, but with CAMS-GLOB-BIOv3.1 (Sindelarova et al., 2022) as a priori biogenic emission inventory | 4.1 | 12.6 | 2.7 | 3.6 | 16.9 | 13.9 |
| S3 | as S0, but with GFED4s (van der Werf et al., 2017) as a priori biomass burning emission inventory | 7.9 | 18.6 | 1.3 | 2.2 | 16.9 | 13.1 |
| S4 | as S0, but with biomass burning emission factors from Akagi et al. (2011) | 7.9 | 17.8 | 2.0 | 2.4 | 16.9 | 13.1 |
| S5 | as S0, but with lower errors on the emission parameters (1.5 times lower) | 7.9 | 17.0 | 2.7 | 3.3 | 16.9 | 14.5 |
| S6 | as S0, but with higher errors on the emission parameters (1.5 times higher) | 7.9 | 18.6 | 2.7 | 3.4 | 16.9 | 11.5 |
| S7 | as S0, but with halved decorrelation lengths for a priori errors on biogenic and fire emissions (100 km instead of 200 km) | 7.9 | 16.8 | 2.7 | 3.4 | 16.9 | 13.7 |
| S8 | as S0, but with doubled decorrelation lengths for a priori errors on biogenic and fire emissions (400 km instead of 200 km) | 7.9 | 18.6 | 2.7 | 2.7 | 16.9 | 13.3 |
| S9 | as S0, but with lower isomerization rates of isoprene peroxy radicals, consistent with Wennberg et al. (2018) | 7.9 | 17.2 | 2.7 | 3.3 | 16.9 | 13.1 |
| S10 | as S0, but using monthly instead of weekly-averaged columns and optimizing monthly instead of weekly emission parameters | 7.9 | 17.1 | 2.7 | 3.8 | 16.9 | 14.9 |