# Peer review of "Weekly-derived top-down VOC fluxes over Europe from TROPOMI HCHO data in 2018–2021"

_EGUsphere, 2023_

## Author Comment (AC1)

**Reply to referee #1**

We thank the anonymous referee for the positive evaluation of the paper. We have made changes to the manuscript based on the recommendations of the referee. The comments below require additional explanations included in this response.

**Lines 64-68: I personally think that saying a particular emission estimate is higher/lower than the MEGAN inventory is not very useful, since (as the authors also pointed out) BVOC emissions estimated from MEGAN can be very different depending on the input PFT data, meteorological data, year of simulation, and also model resolution. I would strongly recommend that the authors revise these statements to give numbers, which would also give the readers a feel of the uncertainty in top-down/bottom-up BVOC emission estimates over Europe.**

This is a good suggestion, though it is important to note that the studies referred to in the text cover different domain sizes and some do not report total emission fluxes. Consequently, intercomparing these different studies is not straightforward. We have included the numbers where available and made additional comparisons in Sect. 5.2 when describing the top-down emission results.

**Section 2 and lines 335-340: Because a strict cloud filtering was applied, the HCHO column dataset used for the inversion may have a 'clear-sky bias', i.e., it is biased high because it only sampled days when the cloud cover was very low. Did the authors consider this when doing the bias correction using the ground-based measurements and when performing the inversion? If so, how? Was the optimization only applied to simulated days/grids with no cloud cover? Did the final weekly emission inversion accounted for the effects of cloud cover?**

When computing the bias correction, only clear-sky data are used for the comparison. This is because the FTIR technique inherently requires clear-sky conditions. This implies that the validity of the bias correction for cloudy conditions is unknown. In principle this is fine, since we apply a strict cloud filter (cloud fraction CF<0.2) to the TROPOMI HCHO data used in the inversion.

However, as the referee highlights, the strict cloud filter might introduce a clear-sky bias in the inversion. The model is sampled at the time of TROPOMI overpass on days with valid TROPOMI measurements, so in general will also be sampled on clear-sky conditions. However, emissions are updated over the full week, hence the emissions are modified during both clear-sky and cloudy conditions even though the inversion only considers clear-sky conditions.

In order to evaluate the impact of the strict cloud filter on the emission inversion, we have conducted a new sensitivity inversion (S11, see Table 3) without this additional cloud filter for the TROPOMI HCHO data. We maintain the standard requirement of QA value > 0.5, which only excludes scenes with >40% cloud cover. In other words, in this sensitivity run, the cloud filter threshold is adapted from 20% to 40%. Because of the change in observational data, we have recalculated the bias correction using unfiltered TROPOMI data and use these bias-corrected unfiltered TROPOMI HCHO data as constraints for the sensitivity run.

In general, we find that the absence of the strict cloud filter leads to slightly lower HCHO columns over land and slightly higher columns over oceans (that is, after applying the new bias correction). In southern

Europe, the change in the bias corrected columns is the smallest. In central Europe, and particularly France, Belgium, the Netherlands, and Germany, the column change is larger, i.e. the unfiltered columns are lower than the standard run by about $1 \times 10^{15}$ molec. cm$^{-2}$. The sensitivity run consequently shows almost no changes in the emissions in southern Europe, and slightly lower biogenic and anthropogenic emissions at higher latitudes, with respect to the standard inversion. In comparison with the standard inversion results, the total change in emissions over the domain is very small. The total isoprene emissions of the S11 sensitivity run are only 0.04 Tg/yr lower than in the standard S0 run (on a total of 18.3 Tg/yr). The anthropogenic emissions are about 0.17 Tg/yr lower in the S11 run. To summarize, the application of the strict cloud filter has little impact on the inversion results. A brief discussion on this sensitivity run is added to the paper. Table 3 and the bar chart in Fig. 11 have also been adapted.

---

## Author Comment (AC2)

**Reply to referee #2**

We thank the referee for the valuable comments that have improved the paper. We have applied all technical corrections and minor changes to the text as pointed out by the referee. Below, we provide additional explanation to main and specific comments.

**HCHO column validation: the authors use FTIR measurements in Europe to validate and propose a bias correction to the HCHO columns to be used in the inversion. Most of the FTIR stations are in the northern part of the domain where emissions and columns are smaller, it may introduce a bias in the bias correction for the large columns, not well represented in the north. The authors stress the lack of ground-based measurement in Southern Europe and its potential impacts in their discussion, but I wonder if they could use FTIR stations in other part of the world, where HCHO columns are higher and similar to the columns in Southern Europe to complement their validation and ensure they do not introduce a bias in the correction.**

The use of FTIR stations outside Europe is an interesting suggestion, but there is no clear best way to address the issue. The bias relation derived in Vigouroux et al. (2020) was based on FTIR data from 25 stations distributed worldwide, but it is not clear whether the additional stations outside Europe are representative for southern Europe. More importantly, the relation derived in Vigouroux et al. (2020) using all FTIR stations globally has a similar regression to that derived in our work. Their slope is slightly larger, which would lead to higher columns in southern Europe and therefore larger biogenic emissions.

**Model evaluation: the comparison with MAX-DOAS measurements and optimized HCHO columns does not show any improvement (except in Greece). The authors state that this might be due to measurements of the MAX-DOAS instruments during cloudy days. Did they try to consider only clear-sky days in their comparison to quantify the impact? It might be interesting.**

This is an excellent suggestion, which led us to investigate the effect of cloud filtering on the comparison with MAX-DOAS observations. In fact, using clear-sky conditions generally does not improve the comparison. More specifically, we find that for several MAX-DOAS stations, the HCHO columns of the optimized model remain lower than in the a priori model, leading to a worsening of the comparison with observations. In general, our analysis shows that the TROPOMI HCHO columns are lower than the a priori model in the Benelux and Germany below 50 degrees north, leading to a decrease of the emissions. Conversely, the observed MAX-DOAS HCHO columns are larger than the a priori model, leading to the aforementioned worsening of the comparison. However, the MAX-DOAS and bias-corrected TROPOMI columns are in relatively good agreement at the German stations, as shown in Fig. S2. This probably points to an issue with the vertical profile in the model, i.e. a likely overestimation of modelled HCHO concentrations at higher altitudes, where TROPOMI is much more sensitive than MAX-DOAS. This would explain the a priori model overestimation with respect to TROPOMI (and not to MAX-DOAS). The inversion leads to a reduction of the emissions to compensate, which decreases the concentrations primarily at lower altitudes, whereas the higher levels are less impacted due to the larger role of background HCHO away from the emission sources.

For the German FTIR stations, we generally see an increase of the emissions. Garmisch and Karlsruhe are located in regions in which the a priori model is lower than TROPOMI, hence we see an increase in the emissions and an improvement in the comparison at those locations. The Bremen FTIR data are poorly sampled, and also reflect days with high biogenic emissions. The MAX-DOAS stations on the other hand

are located in areas in which the observed HCHO columns from TROPOMI are in general lower than the model. This is the case for Bremen, Mainz, Cabauw, De Bilt, Uccle, and to a lesser extent for Heidelberg. For these stations, the vertical profile of the model may be inconsistent with MAX-DOAS and TROPOMI observations, leading to the lack of improvement in the validation results shown in the paper.

To summarize, the data sampling is not the main reason for the lack of improvement of the model with respect to MAX-DOAS data. Rather, all stations (except Athens) are located in a region in which TROPOMI columns are on average similar to or lower than the a priori model, while the MAX-DOAS columns are on average higher than the a priori model. Because MAX-DOAS and TROPOMI probe different altitude ranges, the lack of improvement could be the result of an inconsistent vertical profile of the model. A discussion on this matter has been added to Sect. 5.3.

**Biogenic emissions: in the inversion setup (lines 256-257), it is written that the emission parameters are applied to both isoprene and monoterpene, but the results are only discussed in terms of isoprene emissions in the manuscript. The authors should add a discussion on the monoterpene emissions if they have effectively inversed them or explain why they do not show the results.**

Thanks for this point. The emission increments have indeed been applied to both isoprene and monoterpenes. Monoterpenes have a generally much lower emission flux in our model than isoprene, but can still play an important role in Europe. We have added a brief discussion on monoterpene emissions in Sect. 5.2, and have added the a priori and optimized emission maps in the supplement (as Fig. S7).

**P8, section 2.4: the authors try to use climatological averages based on measurements in the 90s or early 2000s. They mention the difficulties and that the results should be considered with caution. I wonder if something is known or could be derived from climate simulations for example to quantify how far from now these measurements are when considering temperature increase and land-use changes.**

The referee points out that using climatological averages can impact the comparison because the in situ observations are taken over two decades ago. The average temperatures during those years might have been lower than in recent years due to climate change, hence possibly leading to slightly lower HCHO concentrations. However, although the suggestion is interesting, the impact of climate and land-use changes on the concentrations is beyond the scope of this work.

**Inversion setup: It is not completely clear which assimilation window is used. Is it the May-September period?**

The assimilation window consists of May to September, but due to correlations imposed in the emission parameters, emissions are modified throughout the full year as shown in Fig. 7.

**Line 259: On which basis the $10^{10}$ molec.cm$^{-2}$s$^{-1}$ threshold has been chosen? How many pixels are concerned, and where are they located?**

This threshold is chosen arbitrarily, but its low value guarantees that its precise value does not impact the inversion results. The number of pixels varies year to year, but on average it concerns about 400 grid cells. Their locations are primarily coastal regions, in which a very small fraction of land occupies a grid cell leading to low average flux. Additionally, several grid cells in desert regions fall below the threshold.

**Line 282: how have the spatial correlation of anthropogenic emission errors been chosen?**

The description of the spatial correlation of anthropogenic emission errors was indeed incomplete. We inserted the following description:

"Similarly, the anthropogenic emissions from different countries or from different EDGARv4.3.2 categories (road transport, emission processes during production and application, fuel exploitation, energy for buildings, combustion for manufacturing, and the rest) are assumed uncorrelated. This choice is justified by the fact that anthropogenic emission control policies are usually taken at country level and that generally uniform methodologies and emission factors are applied within each country in bottom-up inventories. The spatial correlation coefficient is constant (value of 0.2) for a given category within the same country. No additional dependence on distance is assumed. "

**Line 286 and 570-574: is the filtering out of low columns made on individual observations or after the average?**

This is done after averaging, hence low (and negative) TROPOMI HCHO columns are still taken into account if the average is not below $3\times10^{15}$ molec cm$^{-2}$.

**Line 307-308: Is there a reference that compare the different sensitivity? If yes, it might be interesting to refer to or if not, to provide a plot to illustrate it.**

Such plot is provided in Vigouroux et al. 2020 (Fig. 2) where TROPOMI and FTIR sensitivities are compared. However, we have added a figure as suggested by the referee to show the (average) vertical sensitivities of TROPOMI, FTIR, and MAX-DOAS on the same plot. This figure (shown below) has been added to Sect. 4 as Fig. 3.

[Figure]

*Average vertical sensitivity profiles for TROPOMI (blue), FTIR (orange), and MAX-DOAS (red) instruments. Both the averaging kernels (AVKs) and pressure levels are averaged using all measurements at the station locations.*

**Line 375-376: are the oceanic data excluded with the filter mentioned line 259 or is it a new filter?**

This is a separate filter. The optimization of emissions in specific grid cells is dependent on HCHO columns observations around those grid cells as chemistry and transport effects modify the model columns to a certain distance of the emission area. The first filter (line 259) implies that emissions at a grid cell are not optimized regardless of the HCHO observations around the grid cell. The second filter (line 375) implies that HCHO observations over oceans are not used to optimize emissions over land. That being said, we do see that the HCHO model columns over the Mediterranean Sea and Black Sea are improved by the inversion (see Fig. 4) as a result of emission increases, even though those TROPOMI observations are not assimilated in the inversion.